# The Impact of Japanese Dietary Patterns on Metabolic Dysfunction-Associated Steatotic Liver Disease and Liver Fibrosis

**DOI:** 10.3390/nu16172877

**Published:** 2024-08-28

**Authors:** Takafumi Sasada, Chikara Iino, Satoshi Sato, Tetsuyuki Tateda, Go Igarashi, Kenta Yoshida, Kaori Sawada, Tatsuya Mikami, Shigeyuki Nakaji, Hirotake Sakuraba, Shinsaku Fukuda

**Affiliations:** 1Department of Gastroenterology, Hematology and Clinical immunology, Hirosaki University Graduate School of Medicine, Hirosaki 036-8562, Japan; ssd.tkfm@hirosaki-u.ac.jp (T.S.); satoshis@hirosaki-u.ac.jp (S.S.); tsuyuki31@gmail.com (T.T.); goigarashi19820910@hotmail.co.jp (G.I.); kyoshida@hirosaki-u.ac.jp (K.Y.); hirotake@hirosaki-u.ac.jp (H.S.); sfukuda@hirosaki-u.ac.jp (S.F.); 2Department of Preemptive Medicine, Hirosaki University Graduate School of Medicine, Hirosaki 036-8562, Japan; iwane@hirosaki-u.ac.jp (K.S.); tmika@hirosaki-u.ac.jp (T.M.); nakaji@hirosaki-u.ac.jp (S.N.)

**Keywords:** alpha-tocopherol, antioxidants, body mass index, energy intake, vegetables

## Abstract

This study aimed to investigate the effect of Japanese dietary patterns on metabolic dysfunction-associated steatotic liver disease (MASLD) and liver fibrosis. After excluding factors affecting the diagnosis of hepatic steatosis, 727 adults were analyzed as part of the Health Promotion Project. The dietary patterns of the participants were classified into rice, vegetable, seafood, and sweet based on their daily food intake. Liver stiffness measurements and controlled attenuation parameters were performed using FibroScan. Energy and nutrient intake were calculated using the Brief-type Self-administered Diet History Questionnaire. Univariate and multivariate analyses were used to identify the risk factors for liver fibrosis within the MASLD population. The vegetable group had significantly lower liver fibrosis indicators in the MASLD population than the rice group. The multivariate analysis identified a body mass index ≥ 25 kg/m^2^ (odds ratio [OR], 1.83; 95% confidence interval [CI], 1.01–1.83; *p* = 0.047) and HOMA-IR ≥ 1.6 (OR, 3.18; 95% CI, 1.74–5.78; *p* < 0.001) as risk factors for liver fibrosis, and vegetable group membership was a significant low-risk factor (OR, 0.38; 95% CI, 0.16–0.88; *p* = 0.023). The multivariate analysis of nutrients in low-risk foods revealed high intake of α-tocopherol (OR, 0.74; 95% CI, 0.56–0.99; *p* = 0.039) as a significant low-risk factor for liver fibrosis. This study suggests that a vegetable-based Japanese dietary pattern, through the antioxidant effects of α-tocopherol, may help prevent liver fibrosis in MASLD and the development of MASLD.

## 1. Introduction

Metabolic dysfunction-associated steatotic liver disease (MASLD) is a hepatic phenotype of lifestyle-related diseases with an increasing trend and worldwide prevalence of approximately 30% [1]. In 2023, MASLD was renamed from the previous term, non-alcoholic fatty liver disease (NAFLD), with the diagnostic criteria including the presence of hepatic steatosis along with one or more cardiovascular metabolic risk factors [2]. Hepatic steatosis is often accompanied by cardiovascular risks and can progress to metabolic dysfunction-associated steatohepatitis (MASH), with some cases progressing to liver cirrhosis or hepatocellular carcinoma.

The onset and progression of MASLD is influenced by various factors, among which dietary habits play a crucial role. It is widely known that the Mediterranean diet is effective in preventing NAFLD compared to Western diets [3,4]. The Mediterranean diet is characterized by a higher intake of grains, vegetables, fruits, olive oil, and seafood with a lower consumption of red meat and processed foods than Western diets [5,6,7]. Consequently, the Mediterranean diet, which is plant-based and rich in vegetables, exerts beneficial health effects, such as a reduction in cardiovascular diseases [8]. Similarly, the Japanese diet, which is rich in vegetables, soy products, and seafood and low in meat, resembles the Mediterranean diet in dietary patterns and has been reported to be effective in preventing NAFLD [9,10].

Many studies have investigated the relationship between national dietary patterns, such as Japanese, Mediterranean, and Western diets, and MASLD. However, few epidemiological studies have examined the extent to which differences in dietary patterns within the same region of the country influence the onset and progression of MASLD. The onset and progression of MASLD involves many factors other than diet, such as sex, age, lifestyle, insulin resistance, and lipid metabolism, necessitating adjustment for these confounding factors. In addition, there are no reports investigating the specific forms of Japanese diet that are effective against MASLD and MASH, despite being already known to be effective against NAFLD.

Therefore, we hypothesized that the Japanese diet is effective at preventing MASLD and liver fibrosis. We conducted an epidemiological evaluation involving a large sample of the general population in the rural areas of Japan. This study assessed how the differences in daily Japanese dietary patterns contribute to the onset of MASLD and its progression to liver fibrosis using detailed questionnaires and numerous measurement items. This study aimed to investigate the effect of Japanese dietary patterns on MASLD and liver fibrosis in the general Japanese population.

## 2. Materials and Methods

### 2.1. Study Subjects

This is a cross-sectional study targeting adults aged 20 and over who participated in the Iwaki Health Promotion Project. The data were collected over a 10-day period from 26 May to 4 June 2018. The Iwaki Health Promotion Project is a community-based health promotion initiative targeting residents of the Iwaki area in Hirosaki City, Aomori Prefecture, and is conducted as an annual health checkup every June [11]. All patients participated voluntarily in response to public recruitment. A total of 1056 adults (aged 20–88 years) participated in this study. After exclusion, 329 participants were excluded, and 727 were included in the analysis (Figure 1).

Individuals with the following factors affecting the diagnosis of hepatic steatosis were excluded: patients positive for HBs antigen or HCV antibodies, habitual drinkers (≥30 g/day for men and ≥20 g/day for women), patients taking medications known to cause hepatic steatosis (steroids, methotrexate, amiodarone, and tamoxifen), patients with fewer than 10 Fibroscan measurements, and patients with an interquartile range/median ratio exceeding 0.30, as these were considered unreliable [12].

### 2.2. Transient Elastography

Liver stiffness measurements (LSM) and controlled attenuation parameters (CAP) were performed using FibroScan 530 (Echosens, Paris, France). Both M and XL probes were used, and examinations were conducted by five liver specialists who had received specialized training. LMS and CAP were measured at least 10 times, and the medians were calculated. The relationship between histological liver fat content and CAP values corresponds to 248–267 dB/m for S1, 268~279 dB/m for S2, and 280 dB/m or higher for S3, and a CAP value of ≥248 dB/m was defined as hepatic steatosis [13]. As for liver fibrosis, the LSMs corresponding to histological fibrosis, F0, F1, F2, F3, and F4, are 5.7 kPa, 6.8 kPa, 7.8 kPa, 11.8 kPa, and 25.1 kPa [14]. In other studies, less than 5 kPa of LSM is defined as normal, and in this study, we used an LSM of ≥5 kPa as the cutoff value for liver fibrosis [15].

### 2.3. Clinical Parameters

The following parameters were measured: age, sex, height, weight, body mass index (BMI), aspartate aminotransferase (AST), alanine aminotransferase (ALT), gamma-glutamyl transpeptidase (γGT), fasting blood glucose, hemoglobin A1c (HbA1c), insulin, triglycerides, HDL cholesterol, LDL cholesterol, and smoking and alcohol consumption habits. The Homeostasis Model Assessment of Insulin Resistance (HOMA-IR) was calculated using the following formula: fasting blood glucose (mg/dL) × fasting insulin (μU/mL)/405 [16].

### 2.4. Dietary Pattern Analysis

Energy and nutrient intake were calculated based on the results of the Brief-type Self-administered Diet History Questionnaire (BDHQ), which is a questionnaire developed for large-scale nutritional epidemiological studies. It consists of 80 questions that estimate the intake of 58 food items and over 100 nutrients [17]. It was designed to obtain information on individual nutrient intake, food consumption, and dietary behavior indicators. The BDHQ was sent to the participants in advance and detailed interviews were conducted on the day of the health examination to collect and verify the responses.

To evaluate the Japanese dietary patterns, we conducted principal component analysis (PCA) with fixed lower bounds of 2 and varimax rotation on the 52 food items surveyed via the BDHQ, based on previous reports [18,19]. Participants were then classified into dietary patterns extracted via PCA using nonhierarchical cluster analysis (k-means method). The effects of these dietary patterns on MAFLD and liver fibrosis were compared. For dietary patterns strongly associated with liver fibrosis, we evaluated food items and their components in relation to liver fibrosis.

For the purposes of the univariate and multivariate analysis, MAFLD and liver fibrosis were used as outcome variables, and factors influencing them, which included age ≥ 65 years [20], sex, BMI ≥ 25 [21], smoking [22], exercise habits [23], HOMA-IR > 1.6 [24], HDL cholesterol < 40 [2,25], LDL cholesterol ≥ 140 [2,25], triglycerides ≥ 150 [2,25], and dietary patterns, were used as explanatory variables. For liver fibrosis, an analysis was conducted using specific foods and predominant components in certain dietary patterns as explanatory variables.

### 2.5. Diagnosis of MASLD

Among individuals with hepatic steatosis, participants who met any of the following criteria were diagnosed with MASLD [2]: BMI ≥ 23 kg/m^2^ or waist circumference ≥ 94 cm for men and ≥80 cm for women; fasting blood glucose ≥ 100 mg/dL, postprandial blood glucose ≥ 140 mg/dL, HbA1c ≥ 5.7%, or currently undergoing treatment for type 2 diabetes; blood pressure ≥ 130/85 mmHg or currently undergoing antihypertensive treatment; triglycerides ≥ 150 mg/dL or currently undergoing treatment for dyslipidemia; and HDL cholesterol ≤ 40 mg/dL for men and ≤50 mg/dL for women.

### 2.6. Statistical Analysis

Statistical analyses of clinical data were performed using the Statistical Package for the Social Sciences version 28.0 (SPSS Inc., Chicago, IL, USA) and EZR [26]. Continuous variables were presented as medians and interquartile ranges. The Mann–Whitney U test was used for comparisons between the two groups. Comparisons among the three groups were conducted using the Kruskal–Wallis test and Steel–Dwass multiple comparisons. The relationship between dietary patterns and MASLD incidence was analyzed using univariate analysis. Multivariate analysis was conducted to analyze the dietary patterns, food items, and nutrients related to liver fibrosis in the MASLD population.

The selection of explanatory variables for the multivariate analysis was based on factors that showed significant differences in the univariate analysis for dietary pattern analysis, and factors that were significant in the multivariate analysis of dietary patterns for the analysis of food items and nutrients. Among the explanatory variables in the univariate and multivariate analyses, the Asian standard BMI and HOMA-IR values of 25 kg/m^2^ and 1.6, respectively, were used [21,24]. Statistical significance was set at *p* < 0.05.

## 3. Results

### 3.1. Participant Characteristics

The PCA with varimax rotation extracted four components (Table 1). The cluster analysis was conducted using four factors obtained through PCA, resulting in four groups. The food intake of each group was compared, and each group was named based on the food items that were consumed in significantly larger quantities compared to other groups. The characteristics of the four dietary patterns are presented in Table 2. The first group was named the rice group because it was characterized by a high intake of rice. The second group was named the vegetables group because it was characterized by a high intake of vegetables and mushrooms. The third group was named the seafoods group because it was characterized by a high intake of fish and shellfish. The fourth group was named the sweets group because it was characterized by a high intake of western-style sweets and ice cream. The rice group, which had the largest number of participants and did not exhibit distinct tendencies, had a higher proportion of males, higher age, BMI, CAP value, AST, ALT, γGT, serum glucose, HbA1c, and triglycerides, and lower HDL compared to the vegetable group. However, there were no significant differences to the same extent as observed in the vegetable group between the rice group and the seafood and sweet groups.

### 3.2. Participant Characteristics among the Dietary Patterns in Patients with MASLD

In total, 220 participants met the diagnostic criteria for MASLD. There were no significant differences in the proportion of patients with MASLD among the different dietary patterns (Table 3). Evaluating the characteristics of the MASLD group associated with liver fibrosis across the dietary patterns, the vegetable group had significantly lower AST, ALT, and triglyceride levels than the rice group.

In the univariate analysis of risk factors with MASLD as the outcome, age ≥ 65 years, male sex, BMI ≥ 25 kg/m^2^, smoking habits, HOMA-IR ≥ 1.6, HDL cholesterol < 40 mg/dL, LDL cholesterol ≥ 140 mg/dL, and triglycerides ≥ 150 mg/dL were significant risk factors for MASLD (Table 4). However, the dietary pattern was not identified as a significant risk factor for MASLD.

### 3.3. Risk Factors for Liver Fibrosis in Patients with MASLD

In the MASLD group, 94 patients (42.7%) had liver fibrosis. The univariate analysis identified BMI ≥ 25 kg/m^2^ and HOMA-IR ≥ 1.6 as risk factors for liver fibrosis among patients with MASLD (Table 5). In addition, compared with the rice group, the vegetable group was a low-risk factor for liver fibrosis. In the multivariate analysis, BMI ≥ 25 and HOMA-IR ≥ 1.6 were risk factors for fibrosis, while the vegetable group was a significant low-risk factor.

Investigating the food items and nutrients predominantly consumed by the vegetable group that are associated with reduced liver fibrosis, the multivariate analysis, which adjusted for the fibrosis risk factors BMI ≥ 25 and HOMA-IR ≥ 1.6, identified high intake of carrots, pumpkins, radishes, and turnips as significant food items associated with reduced liver fibrosis (Table 6). Further analysis of the nutrients contained in these low-risk food items revealed that high intake of α-tocopherol was a significant low-risk factor for liver fibrosis (Table 7).

## 4. Discussion

This study revealed that a vegetable-based Japanese dietary pattern was associated with a significantly lower rate of liver fibrosis in patients with MASLD. In addition, it was suggested that a high intake of α-tocopherol, which is found in carrots and pumpkins, is related to the reduction in liver fibrosis rates in the MASLD population.

In this study, we found that the vegetable group had significantly lower liver fibrosis in the MASLD population. The vegetable group of this study represented the dietary pattern of a traditional Japanese diet. The Japanese diet is similar to the Mediterranean diet, rich in vegetables, soybeans, and mushrooms, and is beneficial to the body in the prevention of various diseases such as liver disease and dementia [27,28,29,30,31,32]. Vegetables are rich in dietary fiber, which is converted to butyrate by gut microbiota and believed to inhibit the onset and progression of MASLD via improvement of insulin resistance and lipid abnormalities and anti-inflammatory effects [33,34].

In this study, the vegetable diet group had lower AST, ALT, and triglyceride levels than the other dietary pattern groups in patients with MASLD, which is thought to reflect the hepatoprotective and lipid metabolism-improving effects of the Japanese diet. Previous studies have reported that the Japanese diet reduces the risk of advanced liver fibrosis in patients with MASLD, with the mechanism attributed to muscle and skeletal maintenance through soy consumption [10]. Soy contains isoflavones, which are metabolized by the gut microbiota into equol, a compound with estrogen-like effects that protects against fatty liver [35]. In our study, the vegetable group had a high intake of soy and soy products, suggesting that the high intake of dietary fiber and soy in the vegetable group helped suppress the progression of liver fibrosis in patients with MASLD.

In the analysis of specific food items predominantly consumed by the vegetable group, it was suggested that the intake of α-tocopherol might have been effective in inhibiting liver fibrosis in patients with MASLD. α-tocopherol is a biologically active form of vitamin E with antioxidant properties [36]. The multi-parallel hit hypothesis proposes that not only liver tissue but also other organs, such as the liver, adipose tissue, and intestines, interactively contribute to the onset of MASLD [37]. Oxidative stress, which is known to cause steatohepatitis and liver fibrosis, also plays an important role and is considered a significant pathophysiological factor in MASLD [38,39,40]. Therefore, α-tocopherol, with its antioxidant properties, has been reported to be effective in preventing the onset and progression of NAFLD/NASH [41,42]. Vitamin E has been shown to improve the clinical outcomes in patients with NASH based on real-world practice [43]. Abundant sources of α-tocopherol include nuts, oils, fish, carrots, and pumpkins. This study suggests that the consumption of carrots and pumpkins may inhibit liver fibrosis in patients with MASLD through the antioxidant effects of α-tocopherol.

This study found an association between liver fibrosis and dietary patterns in a population with MASLD, whereas no significant relationship was observed between MASLD incidence and Japanese dietary patterns. Fibrosis involves progression to MASH with accompanying inflammation, which represents the advancement of some cases of MASLD. Factors contributing to MASLD include undernutrition, overnutrition, obesity, glucose metabolism disorders, lipid metabolism disorders, sex, age, oral and gut microbiota, single-nucleotide polymorphisms, and sex hormones, among many other complex interacting factors [41,44,45,46,47]. In this study, differences in age, sex, smoking habits, and exercise habits were observed among the dietary patterns. Therefore, the lack of a significant association between MASLD incidence and dietary patterns may be due to the inability to adjust for other confounding factors aside from the diet of the patients in this study, which was primarily conducted as a health screening project for the general healthy population. The association with dietary patterns was revealed after adjusting for confounding factors and focusing on the analysis of the patients with MASLD. In addition, the study population included a few individuals with a high intake of meat, thus lacking patients with Western dietary patterns. Therefore, since a comparison with meat-based Western diets was not possible, the rice group, which had the largest number of participants among the four dietary patterns, was used as the basis for comparison with other dietary patterns. This study compared other dietary patterns based on the rice group rather than meat-based dietary pattern, which may have contributed to the lack of association between the proportion of MASLD patients and Japanese dietary patterns.

This study has several limitations. First, participants were classified into four groups based on a cluster analysis of food intake, with each group named according to the food items consumed in significantly larger quantities. Consequently, specific cut-off values for dietary intake were not applied. While the observed trends in food consumption offer valuable insights, these findings may not be fully generalizable. Second, the diagnosis of hepatic steatosis was not confirmed using liver biopsy, which is the gold standard for diagnosing hepatic steatosis and fibrosis. However, a liver biopsy is invasive and difficult to perform in the general population. Therefore, we used FibroScan, which is less invasive than liver biopsy. Third, the proportion of patients with fibrosis in the MASLD group was high. Generally, the proportion of MASH with fibrosis within MASLD is 10–20% of the population [48]. The higher proportion of fibrosis (approximately 40%) in this study was attributed to the lower threshold settings for diagnosing fibrosis using LSM. However, when considering the presence of mild fibrosis, this proportion is reasonable. Further studies should be made to address these limitations.

## 5. Conclusions

MASLD is increasing world-wide, and liver fibrosis leads to cirrhosis and hepatocellular carcinoma. This study has demonstrated that a vegetable-based dietary pattern is associated with a lower rate of liver fibrosis in patients with MASLD, which is potentially due to the intake of foods rich in α-tocopherol which has antioxidant properties. These findings suggest that a vegetable-based Japanese diet pattern may improve prognosis by reducing liver fibrosis in MASLD patients.

## Figures and Tables

**Figure 1 nutrients-16-02877-f001:**
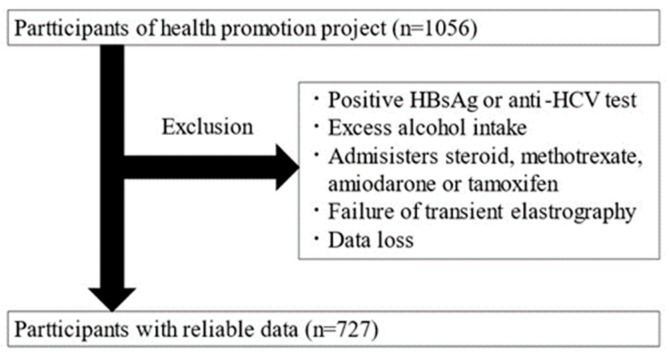
Study enrollment flowchart.

**Table 1 nutrients-16-02877-t001:** Factor loading matrix for dietary patterns identified via the principal component analysis.

	Factor 1	Factor 2	Factor 3	Factor 4
Carrot, pumpkin	0.696	0.046	0.053	0.011
Leafy green vegetables	0.669	−0.068	0.034	0.008
Root vegetables	0.629	0.012	0.041	0.060
Cabbage	0.628	0.119	−0.006	0.050
Mushrooms	0.620	0.080	−0.019	0.055
Raw lettuce, cabbage	0.570	0.055	0.163	0.091
Tofu, fried tofu	0.527	−0.048	−0.063	−0.142
Seaweed	0.517	0.194	−0.183	−0.133
Daikon radish, turnip	0.432	0.244	0.015	0.052
Tomato	0.421	0.172	0.117	0.116
Natto	0.342	0.026	−0.238	−0.195
Cola	−0.336	0.056	0.046	0.006
Mayonnaise	0.334	−0.072	0.320	0.128
Ramen	−0.319	0.079	0.051	0.206
Potato	0.298	0.108	0.033	0.151
Canned tuna	0.273	0.064	−0.04	0.217
Green tea	0.218	0.207	0.064	−0.185
Low-fat milk	0.193	−0.030	0.06	−0.132
Shochu	−0.113	−0.043	−0.047	0.109
Pickled leafy green vegetables	0.102	0.523	−0.075	0.012
Fatty fish	0.288	0.494	−0.127	0.085
Soba noodles	−0.182	0.470	0.006	0.026
Fish with bones	0.156	0.427	−0.172	0.072
Lean fish	0.347	0.419	−0.238	0.074
Dried fish	0.204	0.415	−0.054	0.006
Other pickles	−0.043	0.414	0.020	−0.175
Udon noodles	−0.210	0.372	0.060	0.074
Persimmon, strawberries	0.099	0.371	0.162	−0.063
Squid, octopus, shrimp, shellfish	0.036	0.345	0.011	0.278
Citrus fruits	0.019	0.283	0.069	−0.138
100% juice	0.021	0.162	−0.031	−0.052
Coffee	0.074	−0.109	0.099	0.096
Rice	−0.310	−0.310	−0.689	−0.324
Western-style sweets	−0.049	−0.041	0.580	−0.225
Bread	−0.049	−0.057	0.534	−0.004
Miso soup	0.009	−0.038	−0.485	−0.299
Rice crackers	−0.181	−0.014	0.409	−0.329
Japanese sweets	0.070	0.155	0.361	−0.357
Ice cream	−0.162	−0.065	0.286	−0.034
Sake	−0.034	0.155	−0.206	−0.011
Black tea, Oolong tea	0.015	−0.147	0.173	0.015
Whiskey	−0.077	−0.014	−0.135	0.071
Regular milk	0.054	0.065	0.074	0.063
Pork, beef	0.130	−0.190	0.046	0.533
Ham	−0.045	−0.125	0.142	0.503
Chicken	0.068	−0.027	0.034	0.419
Pasta dishes	−0.157	0.083	0.184	0.344
Beer	−0.056	−0.014	−0.093	0.328
Egg	0.293	−0.240	−0.028	0.319
Other	0.211	0.218	0.187	−0.292
Liver	−0.013	0.227	−0.135	0.261
Wine	0.092	0.074	−0.027	0.174

**Table 2 nutrients-16-02877-t002:** Participants’ characteristics among the dietary patterns.

	Rice Group		Vegetables Group		Seafoods Group		Sweets Group		Rice vs. Vegetables	Rice vs. Seafoods	Rice vs. Sweets
	*n* = 259		*n* = 160		*n* = 151		*n* = 157				
age (year)	52	(39–65)	45	(36–58)	61	(42–70)	55	(39–64)	0.011	0.003	0.981
sex, male	115	(44.4%)	28	(17.5%)	67	(44.4%)	39	(24.8%)	<0.001		
BMI (kg/m^2^)	22.7	(20.5–24.7)	21.5	(19.4–24.1)	22.8	(20.7–25.6)	21.8	(19.3–24.5)	0.007	.0.406	0.018
CAP (dB/m)	222.0	(192.0–261.0)	208.0	(163.8–253.8)	235.0	(192.0–275.0)	213.0	(174.5–262.5)	0.050	0.343	0.477
LSM (kPa)	4.4	(3.4–5.4)	4.2	(3.5–4.9)	4.3	(3.5–5.2)	4.4	(3.6–5.6)	0.849	0.996	0.802
AST (IU/L)	21.0	(17.0–25.0)	19.0	(16.0–22.0)	21.0	(18.0–26.0)	19.0	(16.5–24.0)	<0.001	0.607	0.141
ALT (IU/L)	19.0	(14.0–26.0)	15.0	(12.0–19.8)	19.0	(15.0–25.0)	16.0	(12.0–23.0)	<0.001	0.734	0.058
γGT (IU/L)	21.0	(15.0–32.0)	18.0	(14.0–25.8)	23.0	(17.0–35.0)	18.0	(14.0–30.0)	0.013	0.109	0.232
FPG (mg/dL)	91.0	(86.0–98.0)	88.0	(84.0–93.0)	94.0	(87.0–103.0)	91.0	(85.0–99.5)	0.009	0.003	0.999
HbA1c (%)	5.7	(5.5–5.9)	5.6	(5.4–5.8)	5.7	(5.5–6.0)	5.7	(5.5–5.9)	0.001	0.396	0.954
HOMA-IR	1.1	(0.8–1.6)	1.1	(0.8–1.5)	1.3	(1.0–1.9)	1.2	(0.9–1.6)	0.725	0.004	0.554
TG (mg/dL)	82.0	(54.0–118.0)	62.0	(47.3–84.0)	89.0	(63.0–118.0)	72.0	(53.0–104.0)	<0.001	0.305	0.349
HDL (mg/dL)	60.0	(49.0–71.0)	69.0	(59.0–82.0)	62.0	(52.0–75.0)	64.0	(54.5–77.5)	<0.001	0.273	0.02
LDL (mg/dL)	118.0	(99.0–139.0)	110.5	(92.3–129.0)	119.0	(107.0–142.0)	116.0	(95.5–138.0)	0.053	0.627	0.994
smoking habit (%)	34	(13.1%)	14	(8.8%)	29	(19.2%)	16	(10.2%)	0.031		
exercise habit (%)	30	(11.6%)	38	(23.8%)	37	(24.5%)	22	(14.0%)	<0.001		
MASLD (%)	78	(30.1%)	44	(27.5%)	57	(37.7%)	41	(26.1%)	0.118		

Data are presented as numbers (%) or median (range). BMI, body mass index; CAP, controlled attenuation parameter; LSM, liver stiffness measure; AST, aspartate aminotransferase; ALT, alanine aminotransferase; γGT, γ-glutamyl transpeptidase, FPG, fasting plasma glucose; TG, triglyceride; HbA1c, hemoglobin A1c; HOMA-IR, homeostasis model assessment-insulin resistance; HDL, high-density lipoprotein cholesterol; LDL, low-density lipoprotein cholesterol; MASLD, metabolic dysfunction-associated steatotic liver disease.

**Table 3 nutrients-16-02877-t003:** Participants’ characteristics among the dietary patterns for sample with MASLD.

	Rice Group		Vegetables Group		Seafoods Group		Sweets Group		Rice vs. Vegetables	Rice vs. Seafoods	Rice vs. Sweets
	*n* = 78		*n* = 44		*n* = 57		*n* = 41				
Percentage of MASLD	30.1%		27.5%		37.7%		26.1%		0.118		
LSM ≤ 5 kPa	37	(47.4%)	12	(27.2%)	24	(42.1%)	21	(51.2%)	0.082		
age (year)	57.5	(44.0–66.0)	53.5	(41.3–64.3)	60.0	(44.0–69.0)	60.0	(55.0–68.0)	0.272	0.316	0.102
sex, male	38	(48.7%)	9.0	(20.4%)	30.0	(52.6%)	14.0	(33.3%)	0.019		
BMI (kg/m^2^)	24.9	(23.2–27.7)	24.7	(22.4–27.0)	25.6	(22.7–27.7)	24.9	(23.1–26.3)	0.522	0.801	0.891
CAP (dB/m)	281.0	(262.2–314.3)	281.0	(255.8–299.0)	286.0	(266.0–313.0)	299.0	(268.5–321.0)	0.570	0.732	0.087
LSM (kPa)	4.9	(3.5–6.2)	4.1	(3.5–5.2)	4.6	(4.0–5.6)	5.0	(4.4–6.8)	0.198	0.955	0.115
AST (IU/L)	23.0	(19.0–28.8)	19.5	(16.0–23.0)	22.0	(19.0–28.0)	22.0	(19.0–27.0)	0.002	0.961	0.752
ALT (IU/L)	23.0	(16.0–35.8)	17.0	(14.0–23.3)	23.0	(16.0–37.0)	23.0	(16.0–32.0)	0.004	0.901	0.472
γGT (IU/L)	27.0	(19.0–40.8)	19.5	(17.0–34.0)	29.0	(19.0–41.0)	28.0	(18.0–39.0)	0.106	0.400	0.718
FPG (mg/dL)	95.0	(90.0–108.0)	92.5	(89.0–100.0)	101.0	(93.0–111.0)	95.0	(92.0–110.0)	0.196	0.248	0.767
HbA1c (%)	5.9	(5.6–6.2)	5.7	(5.6–6.1)	5.8	(5.6–6.3)	5.8	(5.7–6.2)	0.386	0.839	0.805
HOMA-IR	1.6	(1.1–2.5)	1.6	(1.1–2.2)	1.8	(1.2–2.6)	1.6	(1.2–2.5)	0.869	0.460	0.595
TG (mg/dL)	102.0	(66.5–152.5)	82.5	(58.5–108.3)	96.0	(81.0–127.0)	102.0	(81.0–149.0)	0.014	0.772	0.889
HDL (mg/dL)	56.0	(46.3–65.8)	61.5	(47.8–69.0)	60.0	(49.0–74.0)	56.0	(47.0–61.0)	0.243	0.161	0.935
LDL (mg/dL)	121.5	(109.0–142.0)	118.0	(99.8–137.0)	120.0	(108.0–136.0)	127.0	(112.0–153.0)	0.197	0.461	0.397
smoking habit (%)	12	(15.4%)	7	(16.0%)	17	(30.0%)	3	(7.1%)	0.034		
exercise habit (%)	11	(14.0%)	10	(22.7%)	14	(24.6%)	6	(14.2%)	0.247		

Data are presented as numbers (%) or median (range). MASLD, metabolic dysfunction-associated steatotic liver disease; BMI, body mass index; CAP, controlled attenuation parameter; LSM, liver stiffness measure; AST, aspartate aminotransferase; ALT, alanine aminotransferase; γGT, glutamyl transpeptidase; FPG, fasting serum glucose; HbA1c, hemoglobin A1c; HOMA-IR, homeostasis model assessment-insulin resistance; TG, triglyceride; HDL, high-density lipoprotein cholesterol; LDL, low-density lipoprotein cholesterol.

**Table 4 nutrients-16-02877-t004:** Univariable analysis of risk factors for MASLD.

	OR	Univariable		*p*-Value
		95%CI		
Age ≥ 65 years	1.94	1.37	2.74	<0.001
Female	0.64	0.46	0.89	0.007
BMI ≥ 25 (kg/m^2^)	7.73	5.29	11.3	<0.001
Smoking habit	1.81	1.16	2.82	0.009
Exercise habit	1.07	0.71	1.62	0.739
HOMA-IR ≥ 1.6	5.79	4.05	8.29	<0.001
HDL cholesterol < 40 (mg/dL)	8.41	3.06	23.1	<0.001
LDL cholesterol ≥ 140 (mg/dL)	1.57	1.09	2.26	0.014
Triglyceride ≥ 150 (mg/dL)	3.92	2.42	6.36	<0.001
Diet pattern				
Rice	1.00			
Vegetables	0.88	0.569	1.36	0.567
Seafoods	1.41	0.922	2.15	0.113
Sweets	0.82	0.526	1.28	0.382

MASLD, metabolic dysfunction-associated steatotic liver disease; BMI, body mass index; HOMA-IR, homeostasis model assessment-insulin resistance; HDL, high-density lipoprotein; LDL, low-density lipoprotein; OR, odds ratio; CI, confidence interval.

**Table 5 nutrients-16-02877-t005:** Univariable and multivariate analysis of risk factors for liver fibrosis in sample with MASLD.

	Univariable				Multivariable			
	OR	95%CI		*p*-Value	OR	95%CI		*p*-Value
Age ≥ 65 years	1.35	0.77	2.35	0.296				
Female	0.85	0.50	1.46	0.558				
BMI ≥ 25 (kg/m^2^)	2.40	1.39	4.15	0.002	1.83	1.01	3.32	0.047
Smoking habit	1.73	0.86	3.46	0.124				
Exercise habit	0.87	0.43	1.75	0.700				
HOMA-IR ≥ 1.6	3.47	1.97	6.09	<0.001	3.18	1.74	5.80	<0.001
HDL cholesterol < 40 (mg/dL)	2.02	0.74	5.53	0.169				
LDL cholesterol ≥ 140 (mg/dL)	0.92	0.51	1.66	0.782				
Triglyceride ≥ 150 (mg/dL)	1.81	0.94	3.49	0.070				
Diet pattern								
Rice	1.00				1.00			
Vegetables	0.42	0.19	0.92	0.030	0.38	0.16	0.88	0.023
Seafoods	0.81	0.41	1.60	0.313	0.66	0.31	1.37	0.260
Sweets	1.16	0.55	2.48	0.700	1.19	0.53	2.67	0.666

MASLD, metabolic dysfunction-associated steatotic liver disease; BMI, body mass index; HOMA-IR, homeostasis model assessment-insulin resistance; HDL, high-density lipoprotein; LDL, low-density lipoprotein; OR, odds ratio; CI, confidence interval.

**Table 6 nutrients-16-02877-t006:** Multivariate analysis of risk factors of food items for liver fibrosis.

	OR	95%CI		*p*-Value
Carrot, pumpkin	0.96	0.92	0.99	0.030
Leafy green vegetables	0.99	0.98	1.01	0.949
Root vegetables	0.99	0.97	1.01	0.459
Cabbage	0.99	0.98	1.01	0.421
Mushrooms	0.97	0.91	1.03	0.309
Raw lettuce, cabbage	0.99	0.97	1.01	0.416
Tofu, fried tofu	0.99	0.97	1.00	0.114
Seaweed	1.01	0.96	1.06	0.643
Daikon radish, turnip	0.94	0.90	0.99	0.012
Tomato	0.99	0.97	1.01	0.414

This multivariate analysis was adjusted for body mass index and homeostasis model assessment-insulin resistance. OR, odds ratio; CI, confidence interval.

**Table 7 nutrients-16-02877-t007:** Multivariate analysis of risk factors of the nutrients for liver fibrosis.

	OR	95%CI		*p*-Value
α-Tocopherol	0.74	0.56	0.99	0.039
β-Carotene	1.00	0.99	1.00	0.118
Retinol	0.99	0.99	1.00	0.125
Dietary fiber	0.94	0.80	1.10	0.519
Vitamin C	0.99	0.98	1.00	0.323
Potassium	0.99	0.99	1.00	0.224

This multivariate analysis was adjusted for body mass index and homeostasis model assessment-insulin resistance. OR, odds ratio; CI, confidence interval.

## Data Availability

The original contributions presented in this study are included in the article. Further inquiries can be directed to the corresponding author.

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
