# Peer review of "The Impact of Japanese Dietary Patterns on Metabolic Dysfunction-Associated Steatotic Liver Disease and Liver Fibrosis"

_nutrients, 2024, doi:10.3390/nu16172877_

Round 1

Reviewer 1 Report

Comments and Suggestions for Authors

1. Line 64-66: Please rewrite the aim of this study. Which “lifestyle-related diseases”? “…prevent their progression, and extend the lifespan in the general Japanese population” are not the aims. 

2. Line 148-149 and Table 2: Please check “….kPa value,…γGTP,…triglycerides…”. Please note the P value of LSM (Rice vs vegetables) is higher than 0.05. Please revise “γGTP” to “γGT”. Please note that the index triglycerides is not included in Table 2.

3. Line 200-209: Please rewrite these two paragraphs to make them more logical.

4. Line 213: which index reflect “the anti-inflammatory effect”?

5. Line 235-237: Please revise to “…between MASLD incidence and Japanese dietary patterns”.

6. Line 250-252: The logic is difficult to understand, so please rewrite.

7. Line 266-268: There were no significant differences in the proportion of patients with MASLD among the different Japanese dietary patterns, so it can not conclude that “Vegetable-based dietary patterns may also prevent MASLD…”.

Comments on the Quality of English Language

Minor editing of English language required.

Author Response

Reviewer 1

We appreciate the time and effort that the reviewers have invested in evaluating our manuscript. We have revised the manuscript according to the reviewers’ suggestions, and our responses to each comment are listed below. The reviewers’ comments are presented in italics.

  • Line 64-66: Please rewrite the aim of this study. Which “lifestyle-related diseases”? “…prevent their progression, and extend the lifespan in the general Japanese population” are not the aims.

Response: Thank you for your comments. As you pointed out, the aim was inappropriate. We corrected the relevant part of the ‘Introduction section’ as follows; This study aimed to investigate the effect of Japanese dietary patterns on MASLD and liver fibrosis in the general Japanese population (Line 64-66).

  • Line 148-149 and Table 2: Please check “….kPa value,…γGTP,…triglycerides…”. Please note the P value of LSM (Rice vs vegetables) is higher than 0.05. Please revise “γGTP” to “γGT”. Please note that the index triglycerides is not included in Table 2.

Response: Thank you for your comments. As you pointed out, the P value of LSM was higher than 0.05. We removed the kPa in the corresponding section of the ‘Result section’ (Line 155). We also revised ‘γGTP’ to ‘γGT’ and added TG in Table 2.

  • Line 200-209: Please rewrite these two paragraphs to make them more logical.

Response: Thank you for your comments. As you pointed out, this sentence was not logical and difficult to understand. We revised the relevant sentence of ‘Discussion section’ as follows; In this study, we found that the vegetable group had significantly lower liver fibrosis in the MASLD population. The vegetable group of this study participants represented the dietary pattern of a traditional Japanese diet. The Japanese diet is similar to the Mediterranean diet, rich in vegetables, soybeans, and mushrooms, and is beneficial to the body in the prevention of various diseases such as liver disease and dementia [27-32]. Vegetables are rich in dietary fiber, which is converted to butyrate by gut microbiota and believed to inhibit the onset and progression of MASLD by improvement of insulin resistance and lipid abnormalities, and anti-inflammatory effects [33, 34] (Line 207-214).

  • Line 213: which index reflect “the anti-inflammatory effect”?

Response: Thank you for your comments. We described the vegetable group as "anti-inflammatory" because AST and ALT levels were lower compared to other dietary patterns. However, this expression was inappropriate because AST and ALT, which are liver enzymes, reflect liver damage. Therefore, we have changed "anti-inflammatory effect" to "low liver damage" (Line 217).

  • Line 235-237: Please revise to “…between MASLD incidence and Japanese dietary patterns”.

Response: Thank you for your comments. As you pointed out, we revised ‘between MASLD incidence and Japanese dietary patterns’ (Line 241).

  • Line 250-252: The logic is difficult to understand, so please rewrite.

Response: Thank you for your comments. As you pointed out this sentence was not logical and difficult to understand. We revised the relevant sentence of ‘Discussion section’ as follows; Therefore, since a comparison with meat-based Western diets was not possible, the rice group, which had the largest number of participants among the four dietary patterns, was used as the basis for comparison with other dietary patterns. This study compared other dietary patterns based on the rice group rather than meat-based dietary pattern, which may have contributed to the lack of association between the proportion of MASLD patients and Japanese dietary patterns (Line 254-259).

  • Line 266-268: There were no significant differences in the proportion of patients with MASLD among the different Japanese dietary patterns, so it can not conclude that “Vegetable-based dietary patterns may also prevent MASLD…”.

Response: Thank you for your comments. As you pointed out, the conclusion was inconsistent with the results of this study. We revised the relevant sentence of ‘Conclusion section’ as follows; MASLD is increasing world-wide, and liver fibrosis leads to cirrhosis and hepatocellular carcinoma. This study suggests that a vegetable-based Japanese diet pattern may improve prognosis by reducing liver fibrosis in MASLD patients (Line 273-276).

Reviewer 2 Report

Comments and Suggestions for Authors

I congratulate the authors for their quality original article  and valuable new data. 

This article almost meets all the conditions necessary for publication in this journal. I suggest few corrections. 

In the section materials and methods under 2.1.- I advise you to specify more clearly the inclusion criteria, the age of the  participants, the type of research and the time period of the research.

If you are able to make a graphical abstract (the article is very appropriate), that would further contribute to the work.  

When calculating the value of steatosis and fibrosis of the liver, was the average value or the median taken? 

I advise you to write reference values ​​for the grades of stetosis (S1, S2, S3) and fibrosis (F0/F1, F2, F3, F4).

Congratulations to the authors.

Author Response

Reviewer 2

We appreciate the time and effort that the reviewers have invested in evaluating our manuscript. We have revised the manuscript according to the reviewers’ suggestions, and our responses to each comment are listed below. The reviewers’ comments are presented in italics.

  • In the section materials and methods under 2.1.- I advise you to specify more clearly the inclusion criteria, the age of the participants, the type of research and the time period of the research.

Response: Thank you for your comments. As you pointed out, we more clearly stated the inclusion criteria. We added the following sentence at the beginning of the ‘Materials and Methods’ section; This is a cross-sectional study targeting adults aged 20 and over who participated in the Iwaki Health Promotion Project. The data was collected over a 10-day period from May 26 to June 4, 2018 (Line 69-71).

  • If you are able to make a graphical abstract (the article is very appropriate), that would further contribute to the work.  

Response: Thank you for your suggestion. We made a graphical abstract.

  • When calculating the value of steatosis and fibrosis of the liver, was the average value or the median taken? 

Response: Thank you for your comments. We used the median in the calculations of steatosis and fibrosis. We added the following sentence to ‘Transient elastography’ within ‘Materials and Methods’ section; LMS and CAP were measured at least 10 times and the medians were calculated (Line 85-86).

  • I advise you to write reference values ​​for the grades of stetosis (S1, S2, S3) and fibrosis (F0/F1, F2, F3, F4).

Response: Thank you for your suggestion. We added the values of Transient elastography corresponding to the histological degree of steatosis (S1, S2, S3) and fibrosis (F0, F1, F2, F3, F4), and the relevant reference (doi: 10.1002/hep.23312). We added the following sentence to ‘Transient elastography’ within ‘Materials and Methods’ section; The relationship between histological liver fat content and CAP values corresponds to 248-267 dB/m for S1, 268~279 dB/m for S2, and 280 dB/m or higher for S3, and a CAP val-ue of >248 dB/m was defined as hepatic steatosis [13]. As for liver fibrosis, the LSMs cor-responding to histological fibrosis, F0, F1, F2, F3, and F4, are 5.7 kPa, 6.8 kPa, 7.8 kPa, 11.8 kPa, and 25.1 kPa [14]. In other studies, less than 5 kPa of LSM is defined as normal, and in this study, we used an LSM of ≥5 kPa as the cutoff value for liver fibrosis [15]. (Line86-92). We also added one additional reference and corrected the reference number accordingly.

Round 2

Reviewer 1 Report

Comments and Suggestions for Authors

The manuscript has been improved, with just two points outlined below.

1. Line 217: Please revise low liver damage to “hepatoprotective” .

2. Line 271-276: Please revise to MASLD is increasing world-wide, and liver fibrosis leads to cirrhosis and hepatocellular carcinoma. This study has demonstrated that a vegetable-based dietary pattern is associated with a lower rate of liver fibrosis in patients with MASLD, which is potentially due to the intake of foods rich in α-tocopherol that has antioxidant properties. These findings suggest that a vegetable-based Japanese diet pattern may improve prognosis by reducing liver fibrosis in MASLD patients.  

Comments on the Quality of English Language

It is fine.

Author Response

We appreciate the time and effort that the reviewers have invested in evaluating our manuscript. We have revised the manuscript according to the reviewers’ suggestions, and our responses to each comment are listed below. The reviewers’ comments are presented in italics.

  • Line 217: Please revise “low liver damage” to “hepatoprotective” .

Response: Thank you for your comments. As you pointed out, we revised ‘hepatoprotective’ (Line 221).

2) Line 271-276: Please revise to “MASLD is increasing world-wide, and liver fibrosis leads to cirrhosis and hepatocellular carcinoma. This study has demonstrated that a vegetable-based dietary pattern is associated with a lower rate of liver fibrosis in patients with MASLD, which is potentially due to the intake of foods rich in α-tocopherol that has antioxidant properties. These findings suggest that a vegetable-based Japanese diet pattern may improve prognosis by reducing liver fibrosis in MASLD patients.” 

Response: Thank you for your comments. As you pointed out, we revised the relevant sentence of ‘Conclusions section’ as follows; ‘MASLD is increasing world-wide, and liver fibrosis leads to cirrhosis and hepatocellular carcinoma. This study has demonstrated that a vegetable-based dietary pattern is associated with a lower rate of liver fibrosis in patients with MASLD, which is potentially due to the intake of foods rich in α-tocopherol that has antioxidant properties. These findings suggest that a vegetable-based Japanese diet pattern may improve prognosis by reducing liver fibrosis in MASLD patients.’ (Line 275-280).
